# Curvature Graph Network

**Ze Ye**
Department of Biomedical Informatics
Stony Brook University
`ze.ye@stonybrook.edu`

**Kin Sum Liu**
Department of Computer Science
Stony Brook University
`kiliu@cs.stonybrook.edu`

**Tengfei Ma**
IBM Research AI
`Tengfei.Ma1@ibm.com`

**Jie Gao**
Department of Computer Science
Rutgers University
`jg1555@rutgers.edu`

**Chao Chen**
Department of Biomedical Informatics
Stony Brook University
`chao.chen.1@stonybrook.edu`

## Abstract

Graph-structured data is prevalent in many domains. Despite the widely celebrated success of deep neural networks, their power in graph-structured data is yet to be fully explored. We propose a novel network architecture that incorporates advanced graph structural information, specifically, discrete graph curvature, which measures how the neighborhoods of a pair of nodes are structurally related. The curvature of an edge $(x, y)$ is defined by comparing the distance taken to travel from neighbors of $x$ to neighbors of $y$, with the length of edge $(x, y)$. It is a much more descriptive structural measure compared to previously ones that only focus on node specific attributes or limited graph topological information such as degree. Our curvature graph convolution network outperforms state-of-the-art methods on various synthetic and real-world graphs, especially the large and dense ones.

## 1 Introduction

Despite the huge success of deep neural networks, it remains challenging to fully exploit their power on *graph-structured data*, i.e., data whose underlying structure is captured by graphs, e.g., social networks, telecommunication networks, biological networks and brain connectomes. Inspired by the power of convolutional neural networks on image data, convolutional networks have been proposed for graph-structured data. Existing works can be roughly divided into two categories, depending on whether convolution is applied to the spectral or spatial domain.

Spectral approaches (Bruna et al., 2013; Defferrard et al., 2016; Henaff et al., 2015; Veličković et al., 2017) apply convolution to eigen-decomposed graph Laplacians and are generally efficient in both computation and memory. However, the learned convolution filters are graph-specific and cannot generalize to different graphs.

Spatial approaches execute "convolution" directly on the graph and operate on the neighborhood as defined by the graph topology. A spatial method iteratively updates the representation of each graph node by aggregating representations from its neighbors, i.e., adjacent nodes (Xu et al., 2018). Nonlinear transformations are applied to the representation passed from one node to another, called a *message*. These transformations have the same input/output dimension, i.e., the dimension of the node representation. They can be shared and learned across different nodes and even different graphs.

For spatial approaches, it is important to incorporate local structural information of the graph. Node degree has been used to re-parametrize the non-linear transformation of messages (Monti et al., 2017) or as an additional node feature (Hamilton et al., 2017). However, node degree is fairly limited; there can be different graph topologies with the same degree distribution. The limitation is illustrated in

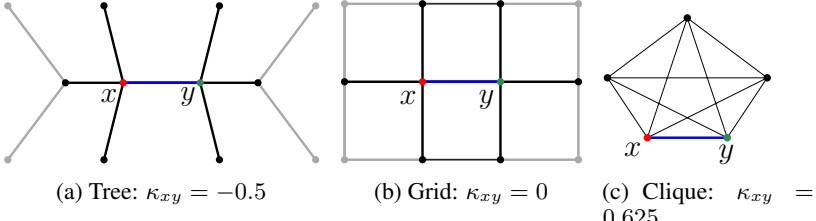

(a) Tree: $\kappa_{xy} = -0.5$      (b) Grid: $\kappa_{xy} = 0$      (c) Clique: $\kappa_{xy} = 0.625$

Figure 1: Illustration of structural information. In all three graphs, the degrees of $x$ and $y$ are the same. However, the Ricci curvature of the edge $(x, y)$ is negative, zero, and positive, respectively. All edges have weight 1. $\alpha = 0.5$ so each node keeps $50\%$ of the probability mass to itself.

Figure 1. Nodes $x$ and $y$ have the same degree in three significantly different graphs: a tree, a grid and a clique. To effectively make use of graph structural knowledge, one would need a feature with more discriminative power; one that can distinguish these three scenarios in Figure 1.

In this paper, we propose a novel graph neural network that exploits advanced structural information. Notice that node degree only describes the number of neighbors of each node, but does not say how these neighbors are connected among themselves. We seek to use structural information characterizing how neighborhoods of a pair of nodes relate to each other. In Figure 1, the neighborhoods of $x$ and $y$ are well separated in different branches of a tree. In a grid graph, the two neighborhoods are within a parallel shift of each other. In a clique, they completely overlap. To quantify such pairwise structural information, we draw inspiration from recent study of *graph curvature* (Ollivier, 2009; Lin et al., 2011; Weber et al., 2016).

Similar to curvature in the continuous domain, e.g., the Ricci curvature of a Riemannian manifold, the discrete graph curvature measures how the geometry of a pair of neighborhoods deviates from a "flat" case, namely, the case of a grid graph. There are several definitions of discrete curvature for graphs. The most notable one is Ollivier's Ricci curvature (Ollivier, 2009). The edges of an infinite grid graph have zero curvature. The curvature of an edge in a tree is negative and is positive in a complete graph. Intuitively, the graph curvature measures how well two neighborhoods are connected and/or overlap with each other. Such information is related to how information propagates in the neighborhood, and should be leveraged by a graph convolutional network.

We propose *Curvature Graph Network (CurvGN)*, the first graph convolutional network built on advanced graph curvature information. The use of curvature information allows CurvGN to adapt to different local structural scenarios and filter messages passed between nodes differently. Notice that curvature captures how easily information flows between two nodes. Within a well-connected community, neighborhoods of adjacent nodes have large overlap and many shortcuts. The corresponding curvature is positive and passing information between the nodes is easy. For an edge serving as a bridge of two clusters, its curvature is negative and it is hard for information to pass. The key to our success is that we choose to be agnostic on whether curvature should be used to block or accelerate the messages in graph convolution. We exploit the curvature information in a data-driven manner and learn how to use it to reweigh different channels of the messages.

To further investigate how curvature information affects graph convolution, we carried out extensive experiments with various synthetic graphs and real-world graphs. Our synthetic graphs are generated according to various well-established graph models, e.g., stochastic block model (Decelle et al., 2011), Watts–Strogatz network (Watts & Strogatz, 1998), Newman–Watts network (Newman & Watts, 1999) and Kleinberg's navigable small world graph (Kleinberg, 2000). On these data, CurvGN outperforms vanilla graph network and networks using node degree information and self attention, demonstrating the benefit of curvature information in graph convolution. Such benefit is more apparent as the graph size increases. We hypothesize that graph convolution alone can adapt to any graph topology, at the cost of more convolutional layers and more training data. This is corroborated by our experiments on real-world graphs. CurvGN outperforms state-of-the-art graph neural networks, especially on large and dense graphs, which tend to have a large variation of local structures. The success of CurvGN demonstrates how topological intuitions inspire better practical solutions. It encourages us to continue the endeavor in applying theoretical insights in successful deployments of deep learning.

## 2 RELATED WORK

We briefly summarize previous works on graph convolution. In early works (Frasconi et al., 1998; Sperduti & Starita, 1997), recursive neural networks were applied on data whose underlying structures are directed acyclic graphs. In Graph Neural Networks (GNNs) (Gori et al., 2005; Scarselli et al., 2009), the recursive network framework was extended to general graphs. Li et al. (2015) introduced gated recurrent units into the framework in order to improve the performance. Since Convolutional Neural Networks (CNNs) have demonstrated strong performance in grid-like-structured data, various methods have been proposed to implement "convolution" on graph-structured data. The efforts can be roughly divided into spectral approaches and spatial approaches. Below we will review both categories in details.

**Spectral approach.** Bruna et al. (2013) transformed the graph convolution into spectral domain multiplication by graph Fourier transform. This method is expensive due to the need to perform matrix eigen-decomposition. Furthermore, it cannot create spatially localized filters as in CNNs. Henaff et al. (2015) applied smooth coefficients on spectral filters to make them spatially localized. Defferrard et al. (2016) used Chebyshev expansion of the graph Laplacian to approximate the filters as a $k$-polynomial function. Kipf & Welling (2016) simplified those methods by reducing to degree one polynomials. Their method is to essentially filter the graph with a kernel whose receptive field is the one-hop neighborhood of each node. The common limitation of spectral approaches is that the convolution filters depend on the Laplacian of each specific graph.

**Spatial approach.** The main challenge of spatial approaches is to design an operator which applies to neighborhoods with different topology and still maintains shared filters. Monti et al. (2017) introduced a mixture model CNN (MoNet) that maps graph neighborhood into spatial neighborhood (with pseudo-coordinates) for spatial convolution. Hamilton et al. (2017) proposed GraphSAGE that samples fixed size neighbors and aggregates their representations. Veličković et al. (2017) proposed Graph Attention Network (GAT), which uses self-attention mechanism to reweigh graph convolution. Recently, there have been other methods which studied graph neural networks from different perspectives, such as pooling (Gao & Ji, 2019; Ying et al., 2018).

**Discrete graph curvature.** Different proposals for discrete graph curvature have been introduced in recent years, including Ollivier-Ricci curvature (Ollivier, 2009; Lin et al., 2011) and Forman curvature (Forman, 2003). We focus on Ollivier-Ricci curvature as it is more geometric in nature, while Forman curvature (Forman, 2003) is combinatorial and is faster to compute. Both curvatures have been applied to real-world graphs. Ni et al. (2015) showed that Ollivier-Ricci curvature can be used to identify backbone edges of an Internet AS graph and is closely related to network vulnerability. Similar applications are shown in the characterization of complex biological networks (Sandhu et al., 2015). There are also applications of Ollivier-Ricci curvature in network alignment (Ni et al., 2018) and community detection (Ni et al., 2019; Sia et al., 2019). Forman curvature is shown to have similar effect (Weber et al., 2017; 2016). To the best of our knowledge, graph curvature has not been used in graph neural networks.

## 3 CURVATURE GRAPH NETWORK

We first formulate the node label prediction problem and explain the mechanism of a Graph Neural Network (GNN). Suppose we have an undirected graph $G = (V, E)$ with features on the vertices $H = (h_1, h_2, \cdots, h_n), h_i \in \mathbb{R}^F$. Here $n = |V|$ is the number of nodes in the graph and $F$ is the feature dimension of each node. Given labels of some nodes in $V$, we would like to predict the labels of the remaining nodes. A GNN iteratively updates the graph $G$'s node representation and eventually predicts node labels. A GNN consists of multiple hidden layers that update node representation from a lower level node representation $H^t \in \mathbb{R}^{n \times F^t}$ where $F^t$ is feature dimension at the $t$-th layer to a higher level representation $H^{t+1} \in \mathbb{R}^{n \times F^{t+1}}$. In particular, $H^0$ is the input feature $H$ and $F^t = F$. The last node representation, $H^T$, are fed to a fully connected layer or a linear classifier to predict node labels. The layers and node representations are illustrated in the top of Figure 2.

Now we explain how to construct hidden layers that update node representations from $H^t$ to $H^{t+1}$. We focus on spatial approaches and treat convolution as a message passing scheme. The $(t + 1)$-th representation of node $x$ is computed by aggregating messages passed from $x$'s neighbors. We also include the message from $x$ to itself. There are several aggregation methods, such as mean, max and sum. We choose summation as it is a commonly used aggregation method (Kipf & Welling, 2016;

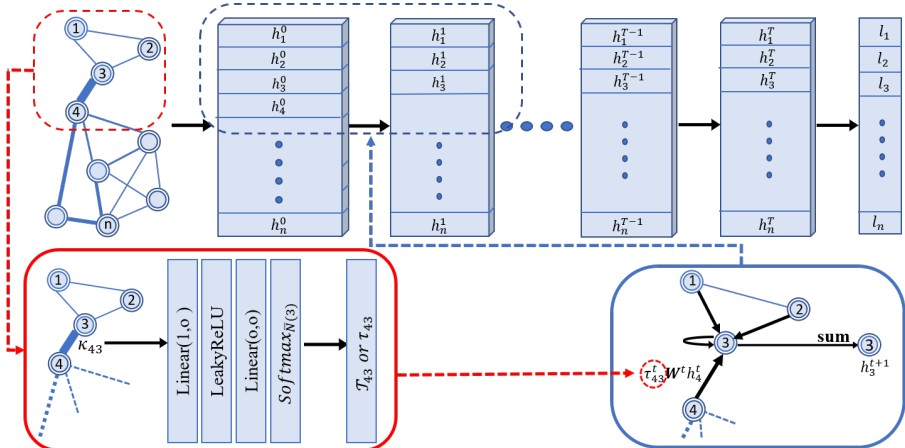

Figure 2: An overview of our Curvature Graph Network. The top row represents the iterative feature update function $H^t$ implemented by consecutive non-linear layers. The bottom row represents the computation of reweighting parameter $\tau_{xy}^t$ from the edge curvature information.

Veličković et al., 2017; Xu et al., 2018). Denote by $\overline{\mathcal{N}}(x) = \mathcal{N}(x) \cup \{x\}$ the neighborhood of $x$ including itself. We have $h_x^{t+1} = \sigma_t \left( \sum_{y \in \overline{\mathcal{N}}(x)} M_{y \to x}^t \right)$, in which $\sigma_t$ is a non-linear transformation. A message passed from $y$ to $x$ is a linear transformation of $y$'s representation. We also introduce a weight $\tau_{xy}^t$ whose purpose will be clear later. Formally, we have $M_{y \to x}^t = \tau_{xy}^t W^t h_y^t$, in which $W^t$ is the linear transformation matrix learned through training. Formally, we have the updating equation

$$h_x^{t+1} = \sigma_t \left( \sum_{y \in \overline{\mathcal{N}}(x)} \tau_{xy}^t W^t h_y^t \right) \tag{3.1}$$

It is crucial to obtain suitable reweighting parameter $\tau_{xy}^t$ since it is directly affecting how information from neighbors of node $x$ is passed to $x$. Some papers use node degree information as $\tau_{xy}^t$ (Kipf & Welling, 2016; Monti et al., 2017) and other work uses joint node features to compute self attention as $\tau_{xy}^t$ (Veličković et al., 2017). We propose to use more advanced structural information, i.e., the Ricci curvature, to compute $\tau_{xy}^t$. It is also known that the reweighting parameter $\tau_{xy}^t$ is not necessarily a scalar. It can be anything between a scalar and a $F^{t+1} \times F^{t+1}$ matrix. Empirically, we choose $\tau_{xy}^t$ as a vector in the experiments because a vector has more expressive power than a scalar and is easier to train than a matrix.

To develop the curvature convolution layer in Equation (3.1), we define discrete Ricci curvature in the context of a general graph in Section 3.1. We explain how to compute $\tau_{xy}^t$ from the curvature in Section 3.2.

## 3.1 RICCI CURVATURE

In Riemannian geometry, curvature measures how a smooth object deviates from being flat, or being straight in the case of a line. Similar concepts can be extended to a non-smooth setting for discrete objects. In particular, curvature has been studied for metric-measure space (Bonciocat, 2014; Bonciocat & Sturm, 2009; Lott & Villani, 2009; Sturm et al., 2006), Markov chain (Ollivier, 2009) and general graphs (Ollivier, 2009; Lin et al., 2011). There are a number of possibilities to define discrete curvature for an edge in a graph. For example, the Gaussian curvature for a triangulated surface could be defined s.t. the curvature of a vertex is $2\pi$ (or $\pi$) minus the sum of corner angles (the angles of the triangles) at it, if it is an interior (or boundary) vertex. The definitions of curvatures easier to generalize in a discrete and general graph setting are sectional curvature and Ricci curvature.

**Curvature computation.** For a point $x$ on the surface, considers two tangent vectors $v$ and $w_x$. Let $y$ be the end point of the tangent vector $v$ at $x$. Imagine transporting in parallel $w_x$ along $v$ to a tangent vector $w_y$ at $y$. If the surface is flat, any pair of points $x'$ and $y'$ which is $\epsilon$ away from $x$ and $y$ along $w_x$ and $w_y$ will have the same distance as $x$ and $y$. Its deviation from $|v|$ defines the sectional curvature. Then averaging it over all directions of $w_x$ gives the Ricci curvature which only depends

on the tangent vector $v$. Intuitively, instead of $w_x$, we can think of $S_x$ be the set of end points of all tangent vectors at $x$ with length $\epsilon$. Again, if we map $S_x$ to $S_y$ using parallel transport along $v$, the distance between a point at $S_x$ and its corresponding image at $S_y$ can be different from $|v|$ if the surface is not flat.

To generalize Ricci curvature to discrete spaces, Ollivier (2009) takes a coarse approach and represents $S_x$ as a probability measure $m_x$ of total mass 1 around $x$. Thus the distance can be measured by Wasserstein distance (or Earth Mover distance) which finds the optimal mass-preserving transportation plan between two probability measures. Then the coarse Ricci curvature $\kappa_{xy}$ on edge $(x, y)$ is defined by comparing the Wasserstein distance $W(m_x, m_y)$ to the distance $d(x, y)$, formally, $\kappa_{xy} = 1 - W(m_x, m_y)/d(x, y)$.

The natural analogy of a small ball $S_x$ at point $x$ in the metric space is the 1-hop neighborhood $N(x)$ of node $x$ in a graph. This motivates the Ollivier-Ricci curvature on graph edges. For an undirected graph $G = (V, E)$, denote the set of neighboring nodes of a node $x \in V$ as $\mathcal{N}(x) = \{x_1, x_2, \ldots, x_k\}$. Then we can define a probability measure $m_x^\alpha$ at $x$:

$$m_x^\alpha(x_i) = \begin{cases} \alpha & \text{if } x_i = x \\ (1 - \alpha)/k & \text{if } x_i \in \mathcal{N}(x) \\ 0 & \text{otherwise} \end{cases}$$

where $\alpha$ is a parameter within $[0, 1]$. It is to keep probability mass of $\alpha$ at node $x$ itself and distribute the rest uniformly over the neighborhood. We set $\alpha = 0.5$, similar to (Ni et al., 2018). To compute the Wasserstein distance $W(m_x^\alpha, m_y^\alpha)$ between the probability measures around two end points $x, y$ of the edge $(x, y)$, the optimal transportation plan can be solved by the following linear programming:

$$\min_M \sum_{i,j} d(x_i, y_j) M(x_i, y_j) \text{ s.t.} \sum_j M(x_i, y_j) = m_x^\alpha(x_i), \forall i; \sum_i M(x_i, y_j) = m_y^\alpha(y_j), \forall j \quad (3.2)$$

where $M(x_i, y_j)$ is the amount of probability mass transported from node $x_i$ to $y_i$ along the shortest path with length $d(x_i, y_j)$. In the language of graph theory, if the Ollivier-Ricci curvature is negative $(W(m_x^\alpha, m_y^\alpha) > d(x, y))$, the edge $(x, y)$ serves as the bridge connecting the neighborhood around $x$ and $y$. This aligns with the intuition in the smooth setting where the small balls $S_x$ and $S_y$ are closer to each other than their centers.

### 3.2 CURVATURE-DRIVEN GRAPH CONVOLUTION

Next we present how Ricci curvature is used in our graph convolutional network. Intuitively, curvature measures how easily a message flows through an edge, and could be used to guide message passing in convolution. But its usage should really depend on the problem and data. We choose to be agnostic on how the knowledge of edge curvature should be used. We resort to a data-driven strategy and learn a mapping function that maps Ricci curvature $\kappa_{xy}$ to the weight of messages, i.e., $\tau_{xy}^t$ in Equation (3.1). We first explain how the mapping is learned end-to-end (CurvGN-1). Next we expand the mapping to a multi-valued version, to incorporate more flexibility in the model (CurvGN-n).

**CurvGN-1.** As mentioned before, $\tau_{xy}^t$ can be anything between a scalar and a $F^{t+1} \times F^{t+1}$ matrix. We first assume $\tau_{xy}^t$ is a scalar. Then the mapping function can be defined as:

$$f^t : \kappa_{xy} \to \tau_{xy}^t \quad (3.3)$$

We create a multi-layer perceptron (MLP) to approximate the mapping function $f^t$ since MLP is proved to be a universal approximation machine and can be easily incorporated into our GNN model for end-to-end training. Denote the MLP at the $t$-th layer as $\text{MLP}^t$. As summation is used as the aggregation function in Equation (3.1), the messages may accumulate to an arbitrarily large value. To prevent a numerical explosion, we apply a softmax function, $S^t$, to $\text{MLP}^t(\kappa_{xy})$ of all neighbors of $x$ including itself, $y \in \overline{\mathcal{N}}(x)$ nodes.

$$\tau_{xy}^t = S^t(\text{MLP}^t(\kappa_{xy})) \quad (3.4)$$

Figure 2 shows how MLP transforms curvature to reweigh messages.

**CurvGN-n.** Messages $M_{y \to x}^t$ are usually multi-channeled. In particular, they are $F^{t+1}$-dimensional. The scalar weight generated using curvature is not necessarily the same for different channels. To

improve the expressing power of $\tau_{xy}^t$, we create a similar mapping function as $f^t$ in Equation (3.3) but it generates a reweighting vector $\mathcal{T}_{xy}^t \in \mathbb{R}^{F^{t+1}}$ instead. In other words, we learn to reweigh different message channels differently. In principle, we can even learn a matrix of dimension $\mathbb{R}^{F^{t+1}} \times \mathbb{R}^{F^{t+1}}$ to reweigh the message. However, a vector has significantly fewer parameters to train and is found to be sufficient in practice.

Using the same strategy as CurvGN-1, the vector $\mathcal{T}_{xy}^t$ is calculated by applying a MLP$^t$ with $F^{t+1}$ outputs. Then, we apply a channel-wise softmax function, $\mathbf{S}^t$, that normalizes the MLP outputs separately on each message channel: $\mathcal{T}_{xy}^t = \mathbf{S}^t(\text{MLP}^t(\kappa_{xy}))$

Substituting $\mathcal{T}_{xy}$ into Equation (3.1), we have the convolution of CurvGN-n:

$$h_x^{t+1} = \sigma_t \left( \sum\nolimits_{y \in \overline{\mathcal{N}}(x)} \text{diag}(\mathcal{T}_{xy}^t) W^t h_y^t \right) \tag{3.5}$$

where $\text{diag}(\mathcal{T}_{xy}^t)$ is a matrix whose diagonal entries are entries of $\mathcal{T}_{xy}^t$. For the details of MLP$^t$, please refer to Appendix A.1.

## 3.3 DESIGN DETAILS OF THE NETWORK

In practice, we use a two-convolutional-layer CurvGN model. The first layer is a linear transform layer that produces an output feature vector paired with a three layer MLP that computes reweighting vector. The output feature is pushed into an exponential linear unit layer to add non-linearity. The second layer is for classification, with the same structure as the first layer except that the output feature is now at length of class number. The hyperparameters are similar to GAT implemented in Veličković et al. (2017). For synthetic experiments, the hidden layer output is reduced to 8 dimensions.

## 4 EXPERIMENTS

We have evaluated our method for both synthetic and real-world graphs. Our method outperforms the state-of-the-art methods, especially on large and dense graphs, which tend to have heterogeneous topological structures. To demonstrate the prediction power, we use different graph theoretical models in synthetic experiments and different parameter settings to gain insights of how curvature information helps graph convolution. We focus on node classification tasks, while our method easily generalizes to graph classification tasks.

### 4.1 GRAPH THEORETICAL MODELS

We generate synthetic data using different graph theoretical models. We start with the Stochastic Block Model (SBM) (Holland et al., 1983), which assumes a partition of the graph into communities. We create random graphs, each with 1000 nodes and equally partition the node set into five disjoint communities. Nodes in the same community have the same class label. Edges are randomly sampled with an intra-community probability, $p$, for nodes within the same community and with an inter-community probability, $q$, for nodes in different communities. We randomly create 100 graphs with $p$ ranging in $\{0.05, 0.07, \cdots, 0.23\}$ and $q$ ranging in $\{0.0, 0.005, \cdots, 0.045\}$. For each generated graph, we randomly select 400 nodes as the training set, another 400 nodes as the validation set and the remaining 200 nodes as the test set. We assign each node with a randomly generated feature of dimension 10 and use them as uninformative input of our CurvGN.

**Baselines.** We compared our proposed methods, **CurvGN-1** and **CurvGN-n**, with two popular state-of-the-art methods: **GCN** (Kipf & Welling, 2016) and **GAT** (Veličković et al., 2017). For these methods, we use the same setting as for Cora dataset mentioned in (Veličković et al., 2017; Kipf & Welling, 2016), except that the dimension of the hidden layer is 8. We also included a baseline method which aggregates messages without reweighting, namely, **Vanilla GN**. Compared with Vanilla GN, GCN reweighs messages using node degrees. GAT reweighs messages using self attention map computed using node representations. CurvGN-1 and CurvGN-n reweigh messages using the scalar and vector forms computed from Ollivier-Ricci curvature.

We ran all methods on 100 random graphs. For each graph, we ran the training and inference task 10 times and took average accuracy. For each training, we run 200 epochs and use validation set for early stopping. Figure 3 shows the heat maps for all methods. The title of each heatmap includes the max and average performance over all parameter settings. In Figure 3(c), we run the same experiments on graphs with different sizes and report the average accuracy.

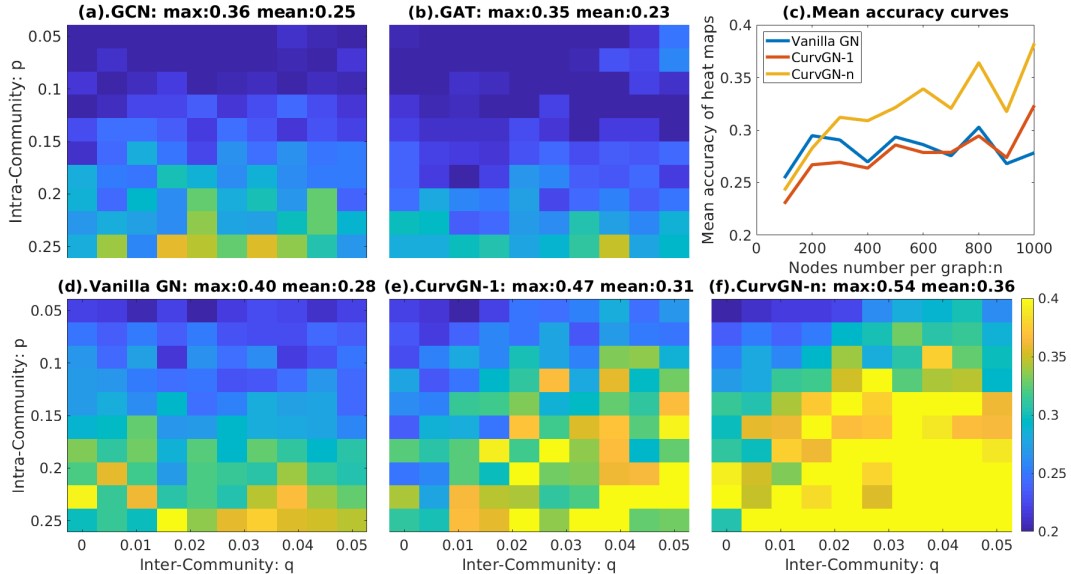

Figure 3: Heat maps on synthetic data by SBM. The top-right figure is the performance of Vanilla GN, CurvGN-1 and CurvGN-n over different graph sizes.

**Discussion.** From the heatmaps, we observe that Vanilla GN, GCN and GAT are not better than random guessing. This implies node degree information is not enough. Meanwhile, we observe improved performance by CurvGN-1, confirming the power of reweighting with curvature in graph convolution. In addition, CurvGN-n outperforms CurvGN-1, suggesting that the multi-channel reweighting based on curvature is beneficial. Furthermore, in Figure 3(c), we observe that the benefit of curvature increases as the graph size increases. *We hypothesize that the graph convolution is sufficient in small graph setting to fully explore the graph structure and advanced structural information becomes important on graphs large in size and rich in structural diversity.*

We also visualize the prediction results on one particular graph generated at $(p, q) = (0.21, 0.025)$ in Figure 4. We observe that CurvGN-1 and CurvGN-n make high quality predictions except for a small portion of data in a few communities. Meanwhile, other baselines can completely mix different communities and results are unsatisfactory.

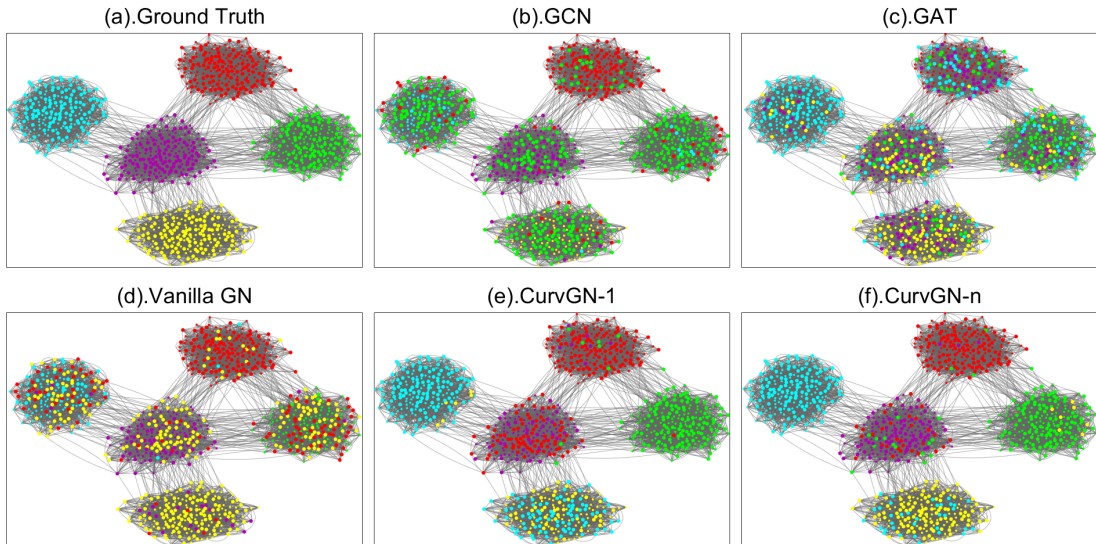

Figure 4: One SBM result with $(p, q) = (0.21, 0.025)$. Small nodes are training set. Large nodes are testing set. Colors correspond to the predicted node labels.

**Further studies on different graph theoretic models**. To further examine the expressive power of Ricci curvature, we run synthetic experiments using several well-accepted graph theoretical models: Watts–Strogatz model (Watts & Strogatz, 1998), Newman–Watts model (Newman & Watts, 1999) and Kleinberg's navigable small world graphs (Kleinberg, 2000). A Watts-Strogatz network randomly rewires edges of a ring graph. A Newman-Watts network randomly adds new edges to the ring. The Kleinberg's model also adds random edges, but the probability of a new edge is inversely proportional to the shortest path distance on the ring. In all these settings, we partition the ring into different communities and design experiments similar to the Stochastic Block Model. More details can be found in the Appendix A.2.

We compared all five methods on these different graph models and report the average accuracy over different parameter settings in Table 1. The standard deviations on these results are generally large as we are averaging over different parameter settings. We observe consistently better performance of CurvGN-n than other methods. This confirms that curvature information is beneficial in a broad spectrum of graphs. We also note that GCN and GAT are indeed doing well for Watts-Strogatz and Newman-Watts models. This is because, in these networks, edge rewiring and addition enlarge differences in node degrees. Bridges crossing different communities tend to have higher node degree. Therefore, node degrees carry useful structural information and can help with graph convolution. We do not observe the same benefit of node degree information in SBM and Kleinberg's model as in these models, node degree is not correlated with the locations of bridges. More heatmap results can be found in the Appendix A.2.

Table 1: Average prediction accuracy on four different graph models.

| Method | SBM | Watts-Strogatz | Newman-Watts | Kleinberg |
|--------|-----|----------------|--------------|-----------|
| **GCN** | 24% | 32% | 32% | 23% |
| **GAT** | 23% | 30% | 30% | 22% |
| **Vanilla GN** | 28% | 26% | 27% | 28% |
| **CurvGN-1** | 31% | 29% | 29% | 27% |
| **CurvGN-n** | **36%** | **32%** | **33%** | **31%** |

## 4.2 REAL-WORLD BENCHMARKS

Our real-world benchmarks include two families of datasets: small sparse graphs and large dense graphs. We compare our networks CurvGN-1 and CurvGN-n with several strong baselines. Besides GCN and GAT that have been used in the experiments on synthetic data, we also compared CurvGN-1 and CurvGN-n with multilayer perceptron (MLP), MoNet (Monti et al., 2017), WSCN (Morris et al., 2019) and GraphSAGE with mean aggregation (GS-mean) (Hamilton et al., 2017). Our method is on par with state-of-the-art methods on relatively small graphs and greatly outperforms state-of-the-art methods on large and dense graphs, which tend to have heterogeneous graph topology.

**Datasets.** We use three popular citation network benchmark datasets: Cora, citeseer and PubMed (Sen et al., 2008). We categorize Cora and citetseer into the first family since both Cora and citetseer graphs are relatively small and sparse. They have thousands of nodes and edges with an average node degree below 2. We also use four extra datasets: Coauthor CS and Coauthor Physics which are co-authorship graphs based on the Microsoft Academic Graph from the KDD Cup 2016 challenge; Amazon Computers and Amazon Photos which are segments of the Amazon co-purchase graph in McAuley et al. (2015). These graphs, together with PubMed, are large and dense graphs. Those graphs have more than 10 thousands node and 200 thousands edges with an average node degree as high as 20. We use the exact data splitting as in semi-supervised learning setting used in Kipf & Welling (2016); Veličković et al. (2017): using 20 nodes per class for training, 500 nodes for validation and 1000 nodes for testing. The descriptions and statistics for all datasets in our experiments can be found in the Appendix A.3.

During the training stage, we set $L_2$ regularization with $\lambda = 0.0005$ for all datasets. Also, all the models are initialized by Glorot initialization and trained by minimizing cross-entropy loss using Adam SGD optimizer with learning rate 0.005. We apply an early stopping strategy based on the validation set's accuracy with a patience of 100 epochs. We compute curvature exactly following Eq. (3.2) for all datasets but one. For the Amazon Computer dataset, we use an approximation scheme

for computational efficiency (Ni et al., 2018). The linear programming problem is solved using an interior point solver (ECOS).

We report the mean and standard deviation of classification accuracy on test nodes on 100 runs and re-use the metrics reported by Monti et al. (2017); Shchur et al. (2018); Veličković et al. (2017) for other state-of-the-art methods. The results are reported in Table 2. Our method is on par with state-of-the-art methods for relatively small graphs and achieves superior performance on large and dense graphs. This is consistent with our conclusion from synthetic experiments: when the graph is large and has heterogeneous topology, advanced structural information becomes critical in graph convolution.

Table 2: Performance on real-world benchmarks.

| Method | Cora | CiteSeer | PubMed | Coauthor CS | Coauthor Physics | Amazon Computer | Amazon Photo |
|---|---|---|---|---|---|---|---|
| **MLP** | 58.2 | 59.1 | 70.0±2.1 | 88.3±0.7 | 88.9±1.1 | 44.9±5.8 | 69.6±3.8 |
| **MoNet** | 81.7 | 71.2 | 78.6±2.3 | 90.8±0.6 | 92.5±0.9 | 83.5±2.2 | 91.2±1.3 |
| **GS-mean** | 79.2 | 71.2 | 77.4±2.2 | 91.3±2.8 | 93.0±0.8 | 82.4±1.8 | 91.4±1.3 |
| **WSCN** | 78.9±0.9 | 67.4±0.8 | 78.1±0.6 | 89.1±0.7 | 90.7±0.9 | 67.6±3.7 | 82.1±1.2 |
| **GCN** | 81.5±0.5 | 70.9±0.5 | 79.0±0.3 | 91.1±0.5 | 92.8±1.0 | 82.6±2.4 | 91.2±1.2 |
| **GAT** | **83.0**±0.7 | **72.5**±0.7 | 79.0±0.3 | 90.5±0.6 | 92.5±0.9 | 78.0±19.0 | 85.1±20.3 |
| **CurvGN-1** | 82.6±0.6 | 71.5±0.8 | 78.8±0.6 | **92.9**±0.4 | 94.1±0.3 | 86.3±0.7 | **92.5**±0.5 |
| **CurvGN-n** | 82.7±0.7 | 72.1±0.6 | **79.2**±0.5 | 92.8±0.3 | **94.3**±0.2 | **86.5**±0.7 | **92.5**±0.5 |

## 5 CONCLUSION

We introduce a novel graph convolution network to leverage advanced graph structural information, namely, the graph curvature. The curvature information effectively helps to achieve superior performance on synthetic and real-world datasets, especially on large and dense graphs. This shows how principled mathematics and theory help the deployment of deep learning and encourages us to continue the endeavor in bridging the gap between graph theoretical foundation and neural networks.

**Acknowledgement.** Liu and Gao would like to acknowledge supports from NSF CNS-1618391, DMS-1737812, OAC-1939459. Ye and Chen's research was partially supported by NSF IIS-1909038, IIS-1855759, CCF-1855760.

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

# A  APPENDIX

## A.1  THE DETAILS OF CURVATURE TRANSFORMATION

We describe the details of $\text{MLP}^t$ for the $t$-th convolutional layer, which maps the curvature $\kappa_{xy}$ of the edge to the weight vector $\mathcal{T}_{xy}^t \in \mathbb{R}^{F^{t+1}}$. $\text{MLP}^t$ has three layers: an input layer, followed by a non-linear transformation layer and an output layer. The input layer **In** linearly transforms the Ricci curvature $\kappa_{xy}$ into an output vector with the same size as the message $M_{y \to x}^t \in \mathbb{R}^{F^{t+1}}$. We use **LeakyReLU** function in our non-linear layer. For output layer **Out**, we use a transformation matrix with size $F^{t+1} \times F^{t+1}$ to compute reweighing vector $\mathcal{T}_{xy}^t$. Formally,

$$\text{MLP}^t = \textbf{Out}(\textbf{LeakyReLU}(\textbf{In})) \tag{A.1}$$

Recall a node also passes a message to itself. To generate the weight vector $\mathcal{T}_{xy}^t$, we set $\kappa_{xx} = 0$, as if the edge $(x, x)$ is a grid edge. For the case when we reweigh the message using a single scalar $\tau_{xy}$ (e.g., CurvGN-1 network), we change the size of transformation matrix of output layer into $F^{t+1} \times 1$.

## A.2  DIFFERENT NETWORK MODELS FOR SYNTHETIC EXPERIMENTS

All three models, Watts-Strogatz, Newman-Watts and Kleinberg's model, are created by randomly modifying/adding edges to a ring graph.[1] A ring graph has $n$ nodes embedded on a circle, with each node connected to its $k$ nearest neighbors. Figure 5a is an example ring graph with $n = 20$ and $k = 4$. To create communities, we partition the nodes into 5 equal-size sets according to their locations on the circle. In addition, we remove the edges cross different communities. Next we explain how edges of the ring graph are randomly changed for Watts-Strogatz, Newman-Watts and Kleinberg's model, respectively.

**Watts-Strogatz Network.** Watts-Strogatz Network (Watts & Strogatz, 1998) is created by randomly rewiring edges of the ring graph with a predefined probability, $p$. See Figure 5b for an example of Watts-Strogatz network.

In our experiments, we generate 100 random Watts-Strogatz graphs of size $n = 1000$ using different parameter combinations of $k$ and $p$: $k \in \{5, 10, \cdots, 50\}$ and $p \in \{0.02, 0.04, \cdots, 0.2\}$. For each graph, the 5 communities correspond to nodes with 5 different labels. We randomly generate a 10-dimensional feature for each node, as in Stochastic Block Model experiments. The training set is created by randomly sampling 400 nodes in one graph. The validation set and testing set are create in the same way with size 400 and 200, respectively. For each graph, we run the experiment 10 times with 200 epochs each time and report the average.

Figure 6a shows the results of all five methods (GCN, GAT, Vanilla GN, CurvGN-1 and CurvGN-n). We observe that CurvGN-n has the best performance compared with others. It suggests that edge curvature information is crucial in prediction: a rewired edge has a high probability to be a bridge with negative curvature. Curvature information can effectively distinguish bridges and intra-community edges, and therefore help graph convolution. It is also worth mentioning that GCN also has good performance. We hypothesize that this is because rewired edges (likely bridges) tend to have higher degrees on adjacent nodes, and thus can be distinguished using degree information alone.

**Newman-Watts Network.** The Newman-Watts network (Newman & Watts, 1999) is similar to the Watts-Strogatz model except that it adds random edges on the ring graph with probability $p$, instead of rewiring existing edges. We run the experiments in the same setting as Watts-Strogatz model. The results are shown in Figure 6b. We observe similar effects as Watts-Strogatz model.

**Kleinberg's Navigable Small World Graph.** Instead of randomly generating edges with a fixed probability $p$, Kleinberg's model (Kleinberg, 2000) adds a fixed number of additional long-range edges to the ring graph. For each node $u$, add $e_l$ random edges $(u, v)$ with $v$ picked with a probability proportional to $1/d(u, v)$, in which $d(u, v)$ is the distance between $u$ and $v$ in the circle.[2] We slightly modify the original definition by making all edges undirected and removing self-loops. Figure 5c shows an example graph of Kleinberg's model with 100 nodes. We observe much fewer long range (cross-community) connections and more intra-community connections than the other models.

---

[1]Note these models can be built on any $d$-dimensional grid. Ring is a special case when $d = 1$.

[2]In general, the probability could be proportional to $1/d(u, v)^m$. We choose $m$ to be 1.

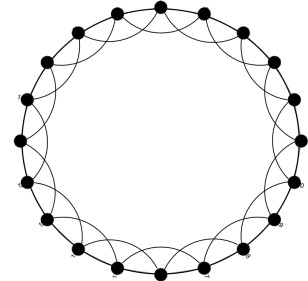

(a) An example ring graph with $n = 20$, $k = 4$.

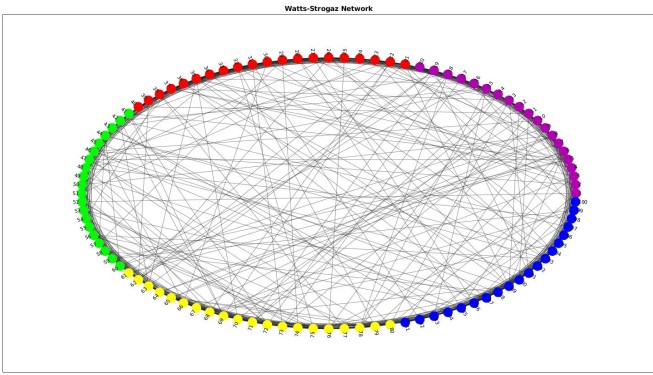

(b) An example Watts-Strogatz graph with 100 nodes.

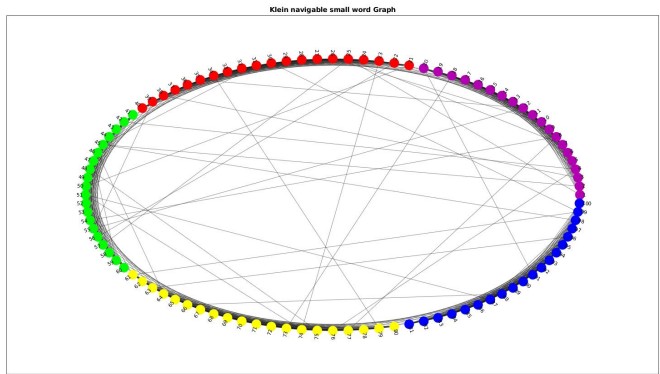

(c) An example Kleinberg's graph with 100 nodes.

Figure 5: Visualization of theoretical graph models

We generate 100 different graphs using different combinations of parameters $e_s$ and $e_l$. Here $e_s \in \{\text{floor}(2.5), \text{floor}(5), \cdots \text{floor}(25)\}$ controls the distance upperbound for short-range neighbors; any nodes within distance $e_s$ of $x$ is connected with $x$. $e_s$ is very similar to $k$ in Wattz-Strogatz graph. Similar to previous models, we run experiments on each graph 10 times and use the validation set for early stopping.

Since the edge is added with a probability proportional to $1/d(u,v)$, there can be long distance inter-community edges. And the node degree can no longer capture the structural information introduced by the added edges. The GCN behaves similar to random guessing in this case. However, the Ricci Curvature is still negative on those inter community edges and it can still detect communities. Figure 6c shows the heatmaps of five different algorithms. CurvGN-n outperforms other methods by a large margin.

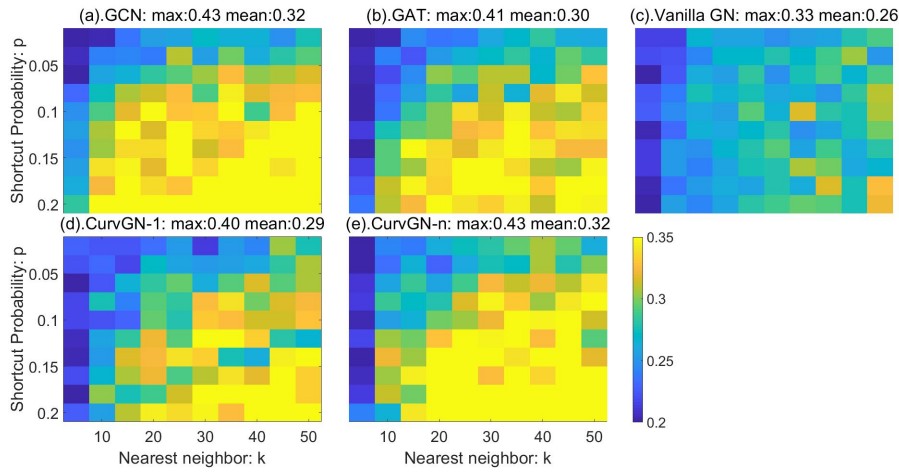

(a) Heatmaps of Watts-Strogatz Network

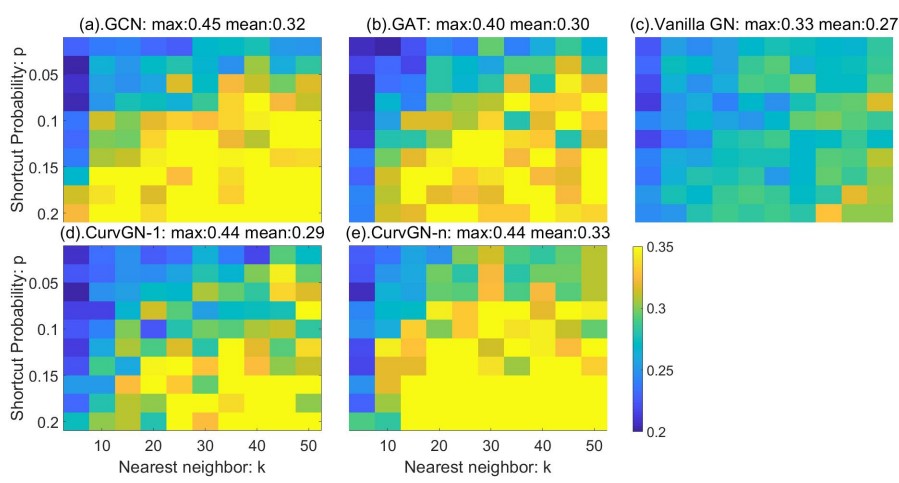

(b) Heatmaps of Newman-Watts Network

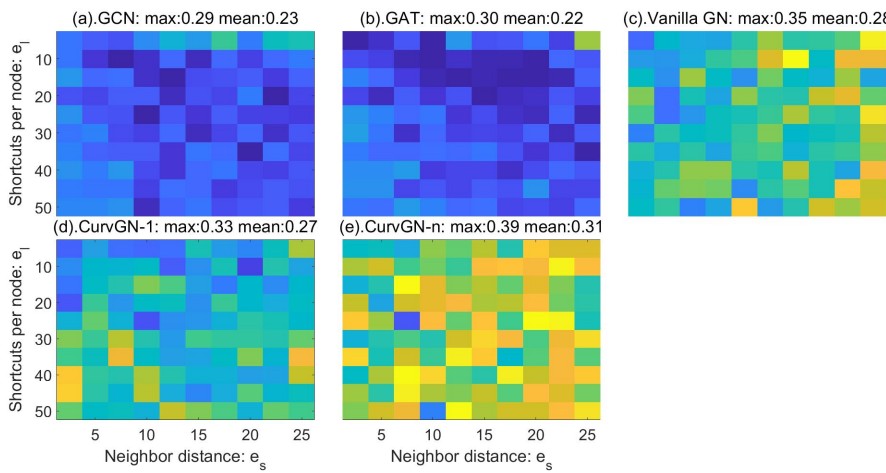

(c) Heatmaps of Kleinberg's navigable small world graph

Figure 6: Results on different theoretical graph models.

Table 3: Statistics of graph datasets including graph characteristics, training details and time for curvature computation. "*" indicates the usage of approximation method.

|  | Cora | CiteSeer | PubMed | Coauthor CS | Coauthor Physics | Amazon Computer | Amazon Photo |
|---|---|---|---|---|---|---|---|
| #Node | 2708 | 3327 | 19717 | 18333 | 34493 | 13381 | 7487 |
| #Edge | 5429 | 4732 | 44338 | 100227 | 282455 | 259159 | 126530 |
| #Edge/#Node | 2.0 | 1.42 | 2.25 | 5.47 | 8.19 | 19.37 | 16.90 |
| #Class | 7 | 6 | 3 | 15 | 5 | 10 | 8 |
| #Feature | 1433 | 3703 | 500 | 6805 | 8415 | 767 | 745 |
| #Training | 140 | 120 | 60 | 300 | 100 | 200 | 160 |
| Curvature Comp. Time(s) | 2.38 | 2.04 | 28.8 | 49.5 | 215.2 | 1873.5* | 1237.6 |

### A.3 STATISTICAL DETAIL OF BENCHMARKS

We describe the statistical details of all datasets in Table 3. Cora and Citeseer are considered as small and sparse graphs while PubMed, Coauthors and Amazons are considered as large and dense graphs.

### A.4 COMPLEXITY OF CURVATURE COMPUTATION

Exactly calculating the curvatures is somewhat time-consuming. We need to solve a linear programming (LP) problem for each edge. For an edge $(x, y)$, the LP problem has $d_x \times d_y$ variables and $d_x + d_y$ linear constraints, in which $d_x$ and $d_y$ are the degrees of the nodes $x$ and $y$. Take the interior point solver (ECOS) as an example, the complexity is $O((d_x \times d_y)^w)$, in which $w$ is the exponent of the complexity of matrix multiplication (the best known is $2.373$).

However, in practice, there are many approximation methods to accelerate the computation of optimal transport distance, among which the best known is the Sinkhorn's algorithm (Cuturi, 2013). It is known to have near-linear time complexity (Altschuler et al., 2017). So for small datasets we can keep using the ECOS to get exact solution while for large datasets (e.g. Computers) we may use advanced approximation methods. In Table 3, we show the computation time for the curvatures with two 18-core CPUs. Note that this computation is only needed once for data pre-processing, and the values can be reused in various downstream tasks.

### A.5 NECESSITY OF CURVATURE TRANSFORMATION

The usage of multi-channel $\mathrm{MLP}^t$ is essential in the success of using the curvature information. To illustrate this, in Table 4, we report the performance of our model if we directly use the curvature to weigh messages, i.e., $\tau_{xy} = \kappa_{xy}$ for all layers. We call this baseline **CurvGN-$\kappa$**.

Table 4: Performance on real-world benchmarks using curvature $\kappa_{xy}$ directly to weigh messages.

| Method | Cora | CiteSeer | PubMed | Coauthor CS | Coauthor Physics | Amazon Computer | Amazon Photo |
|---|---|---|---|---|---|---|---|
| **CurvGN-$\kappa$** | 29.8 | 52.7 | 71.2 | 69.6 | 87.3 | 45.8 | 70.2 |

We observe that curvature alone cannot provide the best reweighing scheme for graph convolution. Only learning from the data helps us determine how weights on different channels should be dependent on $\kappa$. We observe weights of different channels are changing very differently w.r.t. $\kappa_{xy}$; some are increasing w.r.t. $\kappa_{xy}$, others are decreasing. $\mathrm{MLP}^t$ is necessary in learning such dependence from data. Using $\mathrm{MLP}^t$ enhances the model flexibility and model capacity. But also note that curvature $\kappa_{xy}$ is the essential input that the $\mathrm{MLP}^t$ relies on.

