# OpenReview forum: "Curvature Graph Network"
_ICLR.cc/2020/Conference — Accept (Poster)_

### Official Review · AnonReviewer1 · 2019-10-22
**Official Blind Review #1**

**Rating:** 6

**Review:**

In this paper, the authors proposed a novel graph neural network (GNN).
The proposed model combines discrete Ricci curvature of graph edge with GNN.
The combination method is simple but effective, which is easy to implement and is compatible with existing GNN models like GCN.
Experimental results show the potentials of the proposed model to node classification.

My main concerns include:
1. The complexity of the proposed model should be discussed. For a graph with V nodes and E edges, it seems that the authors need to solve E linear programming problems. The scale of each problem is decided by the number of neighbors per node. For large-scale graphs with dense connections, the scalability of the proposed method will be a problem. When the GCN models are with adaptive pooling layers, the curvature will change with respect to the layers of the model, and the scalability problem may become more serious.
2. For the graphs with isolated nodes, do we need to calculate the curvature associated with such nodes? If yes, how to calculate?
3. The authors introduce additional parameters to model the weight tau_{xy} from the curvature kappa_{xy}, i.e., Eq. (3.4). I wonder whether we can use kappa_{xy} directly as the weight tau_{xy} or not. It would be nice if the authors can add an experiment for this variant.

Overall, I think the idea of the proposed method is interesting. Introducing Ricci curvature to GCN is a potential method to take advantage of structural information beyond the adjacency matrix. However, the complexity of the proposed method should be analyzed in-depth.


**Experience Assessment:**

I have published one or two papers in this area.

**Review Assessment: Checking Correctness Of Derivations And Theory:**

I carefully checked the derivations and theory.

**Review Assessment: Checking Correctness Of Experiments:**

I assessed the sensibility of the experiments.

**Review Assessment: Thoroughness In Paper Reading:**

I read the paper thoroughly.

---

> ### Author Response · Authors · 2019-11-14
> **Reply to reviewer#1**
>
> Thank you for the constructive comments. Below we address your concerns one-by-one.
>
> 1, The complexity of the method:
> The complexity of calculating the Ricci curvature exactly would require solving |E| LP problems but one could use approximation methods parallel computation which is much faster. Also, the curvature values are only computed once as a preprocessing step. We reported the computation time in the Appendix (A.4) of the revised version of our paper. Please also see the discussion with Reviewer #2.
>
> In this paper, we mainly focused on the problem of node classification tasks in this paper, so the pooling layer is not necessary. For a graph classification task, as pooling would only reduce the size of the graph, we expect that the computation will be of the same scale as in our setting. For graph classification tasks, most graphs are of much smaller sizes; on these graphs, the computation of curvature can be done within seconds.
>
> 2, Isolated nodes:
> The curvature is defined on edges. For isolated nodes, there is no message passing with the rest of the graph. Thus curvature information is not relevant. In general, we run convolution for each connected component of the graph independently, although they share common filter parameters.
>
> 3, Weight using kappa:
> That is a good idea. We added the discussion to Appendix A.5. We report the suggested baseline using the curvature kappa_{xy} as weights for message passing in Table 5. As we observe, curvature alone cannot provide the best reweighting scheme for graph convolution. Only learning from the data helps us determine how weights on different channels should be dependent on kappa. In appendix A.5 (Figure 7), we also show for one example graph how tau_{xy} at different channels change according to kappa_{xy}. We observe weights of different channels are changing very differently w.r.t. kappa_{xy}; some are increasing w.r.t. kappa_{xy}, others are decreasing. MLP is necessary in learning such dependence from data. Using MLP enhances the model flexibility and model capacity. But also note that curvature kappa is the essential input that the MLP relies on.

---

### Official Review · AnonReviewer3 · 2019-10-23
**Official Blind Review #3**

**Rating:** 6

**Review:**

The work presents a graph neural network that incorporates graph curvature.  The proposed model is able to explore the neighborhood structure of each node, by using the curvature of edges in the proposed framework. Extensive experimental results show the efficacy of the proposed framework. I am not familiar with graph curvature. All I can say is the approach is intuitively appealing, the text is well written and easy to follow, even for an outsider. However, I do not know any related works or what to expect from the results. I could not find anything wrong with this paper, but also do not have any intelligent questions to ask.


**Experience Assessment:**

I do not know much about this area.

**Review Assessment: Checking Correctness Of Derivations And Theory:**

I assessed the sensibility of the derivations and theory.

**Review Assessment: Checking Correctness Of Experiments:**

I carefully checked the experiments.

**Review Assessment: Thoroughness In Paper Reading:**

I read the paper at least twice and used my best judgement in assessing the paper.

---

> ### Author Response · Authors · 2019-11-14
> **Reply to Reviewer#3**
>
> Thank you for carefully reviewing the paper.

---

### Official Review · AnonReviewer2 · 2019-10-24
**Official Blind Review #2**

**Rating:** 6

**Review:**

Curvature graph network

This paper proposes a novel network architecture “curvature graph network” that incorporates the Ricci curvature to fully utilize the graph structure. The Ricci curvature can measure the connectivity around an edge. The higher Ricci curvature, the denser connections there. This is very important information especially for extracting hidden clusters in the graph. The proposed curvature driven graph convolution network shows good experimental results on theoretical synthetic data sets and real world benchmark data sets.

I recommend “weak accept” to this paper since this paper proposes an interesting concept and it looks promising.

Pros.

The authors introduce a very useful metric “Ricci curvature”

CurvGN-1 and CurvGN-2 outperforms existing graph convolution algorithms.

Cons.

To run the proposed algorithm, we have to compute the Ricci curvature values for all edges. This can increase the inference time significantly. The authors should discuss the inference time and check it with a proper experiment.

**Experience Assessment:**

I do not know much about this area.

**Review Assessment: Checking Correctness Of Derivations And Theory:**

I did not assess the derivations or theory.

**Review Assessment: Checking Correctness Of Experiments:**

I did not assess the experiments.

**Review Assessment: Thoroughness In Paper Reading:**

I made a quick assessment of this paper.

---

> ### Author Response · Authors · 2019-11-14
> **Reply to Reviewer#2**
>
> Thanks for your constructive comments. We modified the paper and added the discussion about the computation time for curvature values in the Appendix. In short, by using approximation methods (e.g. sinkhorn) and parallel computation it is feasible and practical as a pre-processing phase even for large datasets. Here we show the computation time for all the graphs in different datasets. We also added this discussion in the revised version of our paper (in Appendix, A.4).
>
> Dataset #Cora #Citeseer #PubMed #CS   #Physics #Computers #Photo
> Time (s) #2.38    #2.04         #28.76  #49.51 #215.24 #1873.5         #1237.55

---

### Decision · Program_Chairs · 2019-12-19

**Decision:**

Accept (Poster)

**Comment:**

The paper presents a novel graph convolutional network by integrating the curvature information (based on the concept of Ricci curvature). The key idea is well motivated and the paper is clearly written. Experimental results show that the proposed curvature graph network methods outperform existing graph convolution algorithms. One potential limitation is the computational cost of computing the Ricci curvature, which is discussed in the appendix. Overall, the concept of using curvature in graph convolutional networks seems like a novel and promising idea, and I also recommend acceptance.